# Tunable Optical Properties of Cu/VSe_2_ from the Visible to Terahertz Spectral Range: A First-Principles Study

**DOI:** 10.3390/ijms26062527

**Published:** 2025-03-12

**Authors:** Elaheh Mohebbi, Eleonora Pavoni, Pierluigi Stipa, Luca Pierantoni, Emiliano Laudadio, Davide Mencarelli

**Affiliations:** 1Department of Science and Engineering of Matter, Environment and Urban Planning (SIMAU), Marche Polytechnic University, 60131 Ancona, Italy; e.mohebbi@staff.univpm.it (E.M.); e.pavoni@staff.univpm.it (E.P.); p.stipa@staff.univpm.it (P.S.); 2Information Engineering Department, Marche Polytechnic University, 60131 Ancona, Italy; l.pierantoni@staff.univpm.it (L.P.); d.mencarelli@staff.univpm.it (D.M.)

**Keywords:** VSe_2_, copper, electronic property, optical responses, I(V) characteristics

## Abstract

In this study, Density Functional Theory (DFT) and Density Functional Tight-Binding (DFTB) calculations were used to study two different interfaces of Cu/VSe_2_ as well as four nanodiodes of VSe_2_ bulk including/excluding the Cu layer. We calculated the electronic and optical properties of two systems of two Cu/VSe_2_ in which Cu atoms are positioned on the top and at the corner of the VSe_2_ monolayer lattice. The electronic band structure calculations revealed that the metallic properties of the VSe_2_ monolayer did not change with the interface of Cu atoms; however, the peak around the Fermi level (E_F_) in Cu/VSe_2_(Top) shifted downward to lower energies. The optical properties showed that in the visible range and the wavelengths related to the interband transition/intraband excitation of Cu atoms, the enhancement of Re(ω) values could be observed for both Cu/VSe_2_(Top) and Cu/VSe_2_(Corner) nanostructures, while in infrared/terahertz ranges, less/more negative values of Re(ω) were predicted. Through the effect of Cu atoms on the VSe_2_ monolayer, the intensity of the peaks in the Im(ω) part of the dielectric constant was increased from 0.2 THz for Cu@VSe_2_(Top) and 2.9 THz for Cu@VSe_2_(Corner) instead of the zero constant line in the pure system of VSe_2_. Refractive index (n) calculations indicated the higher indices at 5.4 and 4.6 for Cu/VSe_2_(Top) and Cu@VSe_2_(Corner), respectively, in comparison to the value of 2.9 for VSe_2_. Finally, DFTB calculations predicted higher current values from I(V) characteristic curves of Au/Cu/VSe_2_/Au and Ag/Cu/VSe_2_/Ag nanodiodes concerning two other devices without the presence of the Cu layer.

## 1. Introduction

Two-dimensional material transition-metal dichalcogenides (TMDs), which are built via van der Waals (vdW) epitaxial growth, have been widely investigated due to their remarkable electronic and optical properties [1,2] thanks to their tunable band gap, direct/indirect band gap and efficient optical absorption [3,4,5,6]. Moreover, TMDs have attracted enormous research interest due to their other novel physical properties. Strong excitonic effects arise from the Coulombic interactions between carriers caused by the reduced dielectric screening [6,7,8,9]. Vanadium diselenide (VSe_2_) is a typical TMD that exhibits metallic properties due to the strong electron coupling between all neighboring V^4+^−V^4+^, and the 1T-VSe_2_ polymorph of this material shows an extreme electrical conductivity of 10^6^ S/m [10,11,12]. Importantly, the 3*d*^1^ odd-electronic configuration of V^4+^ ions also provides rich spin-related information. Dai and co-workers used a theoretical approach to report that a pristine 2D monolayer VSe_2_ may exhibit intrinsic magnetic ordering and behave as an unwonted 2D magnetic material [10].

VSe_2_ is a particularly attractive candidate system in which the bulk of VSe_2_ has a significant structural contribution in the out-of-plane direction (Figure 1A), which has been attributed to a nesting of the strongly three-dimensional Fermi surface at the corresponding wave vector [13]. When reducing the material thickness to a single monolayer (Figure 1B), where the electronic structure must become strictly two-dimensional, a major influence on the charge order in VSe_2_ [14] can be expected. Moreover, as mentioned before, first-principles calculations have consistently predicted that a robust ferromagnetic order should emerge for monolayer VSe_2_, with a pronounced exchange splitting of the near-E_F_ electronic states in the 1T polymorph of more than 500 meV [12].

Particular physical properties can provide new ideas for the development of efficient quantum size and localized surface plasmon resonance (LSPR) effects [15,16]. So far, experimental and computational studies have revealed that the LSPR phenomenon of nanomaterial interfaces can extend the light-absorb range of semiconductors from the visible-light to near-infrared-light zone by improving the structures or particle dimensions [17,18] and dielectric surrounding [19,20]. Moreover, the powerful local electric fields produced at the metal–semiconductor interface can remarkably encourage the separation level of photo-generated electron–hole pairs. Elemental metals, i.e., Au, Ag, Cu, Al and Bi, have been explored and found to have LSPR effects, with applications in solar cells [21], photocatalysts [22] and biosensors [23]. The plasmon resonance absorption peak of these metals can be observed around wavelengths of 517 nm (Au), 364 nm (Ag), 675 nm (Cu) and 325 nm (Al). Compared with other noble metals, Cu as a low-cost transition metal has received more attention from developers [24].

Direct energy from light or electricity can be channeled through the decay of surface plasmons, leading to the excitation of hot electron–hole pairs. In metal nanostructures, hot electron generation can take place through intraband excitation within the conduction band edge (CBE) or via interband transitions, which result from transitions from other bands to the unoccupied states of the CBE. It has been observed that in copper (Cu), population decay occurs through the conversion of plasmons into photons (radiation damping) and through non-radiative decay into electron–hole pairs. As illustrated in Figure 2, both intraband excitations and interband transitions in bulk Cu can occur along the x path in the band structure diagram. Due to the requirement that electron energy levels conform to the Fermi–Dirac distribution; incident light excitation leads to the excitation of electrons from occupied states below the Fermi level (E_F_) to unoccupied states above E_F_. Given Cu’s metallic properties, these excited electrons become free carriers that can collectively oscillate in response to alternating external electromagnetic fields. This is the fundamental mechanism behind the LSPR effects produced by Cu [25,26].

Since most previous studies are related to the morphology, band structure and electromagnetic properties of VSe_2_ polymorphs, optical applications are rarely reported for this material. On the other hand, the effects of Cu, Au and Ag metals are mostly considered in the visible region to the infrared zone in the interactions with the semiconductors. In the present work, on the basis of DFT calculations, the regulation rules of Cu/VSe_2_ interfaces were investigated systematically and comprehensively for the terahertz (THz) application. We developed two kinds of modeling based on material characterization and devices as follows: (i) the material interfaces linked into an architecture in which we calculated the electronic and optical properties of Cu/VSe_2_ (top and corner positions of Cu on the surface of VSe_2_) using the accurate Heyd–Scuseria–Ernzerhof (HSE) hybrid methodology; (ii) self-consistent charge Density Functional Tight-Binding (DFTB) calculations carried out to evaluate the electrical features of Cu/VSe_2_ nanodevices using different Au and Ag metallic electrodes. Combinations of these methods create new ideas for the constructive application of VSe_2_-based and other novel 2D materials in low-frequency THz applications.

## 2. Results and Discussion

### 2.1. Structural Parameters and Binding Energy

Bulk VSe_2_ has a layered crystal (space group *p*6_3_*mc*) with separate layers stacked along the (001) direction (c-axis direction in Figure 1A). From our DFT calculations, the lattice constants of a = b = 3.42 Å and c = 6.34 Å were predicted for the bulk VSe_2_. For the VSe_2_ monolayer, the V-Se bonds are 2.51 Å, and the corresponding Se-Se distances are 3.67 Å. The computation results related to structural parameters of the VSe_2_ bulk and monolayer in this study are in good agreement with the experimental observations and theoretical calculations [27,28,29]. According to our previous DFT works based on the adsorption of semiconductor materials of SnSe and SnSe_2_ on graphene [6], in which the effect of the chemical nature of interfaces on metal–semiconductor hybridization has been described, here, we are focusing on metallic–metallic interaction with Cu adsorption at two different sites over the VSe_2_ monolayer. Figure 3 indicates two geometries, where Cu is adsorbed over the VSe_2_ surface, near to the V atom (Cu@VSe_2_ (Top)), or more simply cornered (Cu@VSe_2_ (Corner)), closer to one of the Se atoms on the unit cell of the VSe_2_ monolayer. We initially started by considering the geometrical stability levels of different geometries, and the results showed similar binding energies of −0.123 and −0.118 eV for Cu@VSe_2_(Top) and Cu@VSe_2_(Corner), respectively. 

### 2.2. Electronic Structure Calculations

We thus calculated the band structures, projected density of states (PDOS) and Mulliken charges of VSe_2_ monolayer and interfaced systems. The outcomes revealed that in both interfaces, we had found a zero-band gap. In a comparison between pure VSe_2_ and Cu@VSe_2_(Top) (Figure 4), the band structure of pristine VSe_2_ indicated that the CBE band crossed the Fermi level and, in M-K paths in BZ, had its maximum/minimum level, respectively, while the top of the valence band edge (VBE) was slightly away from the G point. Moreover, by zooming in the band structures in the zone of G-K, we consistently found a saddle point (SP). The difference in the band structure of the VSe_2_ monolayer and Cu/VSe_2_(Top) mostly refers to the band dispersions in the zone of SP, in which, after the interaction of Cu with VSe_2_, the CBE band shifted to lower energies.

To further validate these results, we carried out PDOS calculations of two systems using the screened hybrid HSE06 functional. The projected PDOS of the VSe_2_ monolayer demonstrated that the states between 1 eV and 3.4 eV mainly comprise the V-3*d* orbitals. According to the octahedron crystal field, the V-3*d* orbitals split into the t_2_g and the e_g_ manifolds, which, in the PDOS plot, are contributed at 1–1.4 eV and 1.9–3.4 eV, respectively. Compared with the PDOS of the pristine system in Figure 4A, in the intercalated VSe_2_ area with Cu atoms (Figure 4B), the peak around the Fermi level (E_F_) shifts downward below the E_F_ and nearly exhibits a cutoff in this area. That feature indicates a filling of this band.

As confirmed by the Mulliken charge calculations, adsorbed Cu atoms lost charge, and the VSe_2_ substrate received charge. These results are in good agreement with the electron-withdrawing character of Cu species, which usually leave the substrate positively charged. The magnitude of the charge transfer between Cu, V and Se along with the absorption can support a more comprehensive description of the Cu/VSe_2_ interaction. Mulliken charge calculations reveal a slight transfer of electrons to the d_z_^2^ of V and *p_x_* and *p_y_* orbitals of the Se atom near to Cu. The total charges of *p* orbitals of the Se atom lightly changed from 4.125 in the pristine system to 4.552 in the interfaced system. On the other hand, the charge of d_z_^2^ orbitals of V increased from 0.480 to 0.898. Since d_z_^2^ orbitals are the lowest 3*d* orbital in the CBE band, they can cross the Fermi level and move to lower-energy physics of the VSe_2_ monolayer. A similar PDOS description was produced for Cu/VSe_2_(Corner), and the results showed a negligible difference between the two different adsorption sites. The PDOS behavior in this study showed similarity with those from relevant experimental works in literature [30].

### 2.3. Optical Properties

From a different perspective, Figure 5 and Figure 6 compare the calculated real (Re(ω)) and imaginary (Im(ω)) parts of the dielectric constant of adsorbed systems with respect to pure VSe_2_. Compared to the electronic properties, in which the band structure and PDOS of VSe_2_ are not drastically affected by the presence of Cu atoms, the Cu/VSe_2_ interface can significantly change the optical properties of pristine material. The dielectric functions along the a-, b- and c-axes present in-plane and out-of-plane directions, respectively. The incident in-plane directions are parallel to the symmetrical axis, while the incident out-of-plane direction is perpendicular to the symmetrical axis.

The optical properties of Cu bulk show a wavelength of 675 nm where the LSPR of Cu is expected to occur (intraband excitation), while in wavelengths below 550 nm, interband transitions can be observed (Figure 5A) [26]. As illustrated in Figure 5B–D, in the wavelength of the interband transitions for Cu, increasing Re(ω) was seen from 1.16 in pure VSe_2_ to 1.96 for Cu/VSe_2_(Top) and 1.94 for Cu/VSe_2_(Corner). Similarly, Re(ω) values indicated that the low loss of Re(ω) corresponded to Cu/VSe_2_(Top) (Figure 5C) and Cu/VSe_2_(Corner) (Figure 5D) with respect to pure VSe_2_ at the intraband wavelength. At higher wavelengths near to the infrared region at 700–1000 nm, the Re(ω) of adsorbed systems transitioned to less negative values, while at 2100 nm (142 THz), the maximum negative values of −4.72 and −4.00 for Cu/VSe_2_(Top) and Cu/VSe_2_(Corner) were observed, respectively, in comparison to the positive Re(ω) values in VSe_2_.

Since, in this study, considering the optical properties of Cu@VSe_2_ in the low-frequency THz range is so important, we report the Re(ω) and Im(ω) parts of the dielectric constant as well as the refractive index (n) of pure and interfaced systems in the range of 0–10 THz. The absorption edge of Im(ω), which is equal to the optical gap, is located at zero along all directions for pure VSe_2_ (see Figure 6A). Our HSE results reveal that in both interfaced cases of Cu@VSe_2_(Top) and Cu@VSe_2_(Corner), the intensity of the peaks started to increase from 0.2 THz for Cu@VSe_2_(Top) and 2.9 THz for Cu@VSe_2_(Corner). However, the polarization induced by light showed different results, in which the in-plane (a-) axis indicated higher intensities with respect to the in-plane (b-) and out-of-plane c-directions in Cu@VSe_2_(Top). About the Cu@VSe_2_(Corner) interface, we could not observe significant differences between the a- and b- in-plane axes, while the intensity of b- in-plane was higher than the same axis in the adsorbed system of Cu@VSe_2_(Top).

The Re(ω) part of different pure and adsorbed systems (see Figure 6), which is related to the static dielectric constant, showed the positive values of 8.03 and 2.04 along the in-plane and out-of-plane directions, respectively, for the pristine VSe_2_ monolayer. These values significantly increased when VSe_2_ interacted with Cu atoms by 27.50, 17.45 and 3.69 for Cu@VSe_2_(Top) and 19, 20.38 and 3.68 for Cu@VSe_2_(Corner) along the in-plane (a-), in-plane (b-) and out-of-plane directions, respectively.

Figure 7A–C show the refractive index (n) of pure VSe_2_ (A) in comparison to the interfaced systems of Cu@VSe_2_(Top) (B) and Cu@VSe_2_(Corner) (C). In Figure 7A, the static refractive indices are 2.9 and 1.8 in the in-plane (a- and b-) and out-of-plane directions, respectively, in the range around 1 THz. Moreover, the corresponding values of the refractive indices are at 5.4 and 4.2 for Cu/VSe_2_(Top) and 4.5 and 4.6 for Cu@VSe_2_(Corner) along the in-plane (a-) and in-plane (b-) directions, respectively, while a similar value of 1.9 was predicted along the out-of-plane direction, in both adsorbed systems.

### 2.4. I(V) Characteristics

After characterizing the materials of both pure and Cu-adsorbed VSe_2_ systems, four distinct nanodiodes were modeled, namely Au/VSe_2_/Au, Ag/VSe_2_/Ag, Au/Cu/VSe_2_/Au and Ag/Cu/VSe_2_/Ag. Each diode had identical dimensions of 20 × 20 × 83 Å^3^ along the x, y and z axes. These devices were constructed using Au and Ag electrodes, with the inclusion of a Cu layer on top of the VSe_2_ material in two of the configurations (refer to Figure 8). The rationale for simulating nanodevices with both Au and Ag lies in the fact that Au possesses a higher work function (5.1 eV) compared to Ag (4.2 eV), which may lead to variations in the electrical properties in diodes. I(V) properties were calculated under a zero-gate ambient voltage in different devices.

The findings illustrated in Figure 9 demonstrate the non-linear I(V) characteristics across four nanodiode configurations. The data indicated that the Au/Cu/VSe_2_/Au and Ag/Cu/VSe_2_/Ag diodes exhibited significantly higher current values in both positive and negative voltage ranges from −1 V to 1 V when compared to the devices lacking a copper layer. Additionally, the I(V) characteristics of the Ag/VSe_2_/Ag diode without the presence of Cu atoms showed some noise because of very high resistance behavior. The resistance fluctuation of noise could be observed for some diode samples, showing the I(V) characteristic dependent on the electric field orientation [31]. Indeed, our theoretical results suggest that the Ag/Cu/VSe_2_/Ag device with an appropriate content of Cu possesses abundant electronic traps, which effectively inhibits the recombination of charge carriers and improves the resistivity in I(V) curves of VSe_2_. Nevertheless, the difference in work function between Au and Ag did not lead to substantial changes in the current values for the Au/Cu/VSe_2_/Au and Ag/Cu/VSe_2_/Ag devices, as shown in Figure 9. Only within the voltage range of ±0.3 to ±1 V did the Ag/Cu/VSe_2_/Ag diode show marginally higher currents compared to the Au/Cu/VSe_2_/Au configuration. Lee et al. [32], through experimental and theoretical investigations, illustrated the effectiveness of photovoltaic devices enhanced by electromagnetic fields. Their research highlighted the differences in hot electron generation resulting from intraband excitation versus interband transition in a plasmonic Cu/TiO_2_ nanodiode utilizing Au electrodes, by assessing the current conversion efficiency with a monochromator system.

## 3. Materials and Methods

Density Functional Theory (DFT) [33] calculations were utilized with the Quantum ATK (QATK) code [34]. This software is designed to analyze the chemical and physical properties of various materials and devices [35,36,37]. The Kohn–Sham (KS) expression [33,38] was addressed using the linear combinations of atomic orbitals (LCAO) basis set [39]. To effectively model the valence orbitals that approximate the core electrons for the elements V, Se, Cu, Au and Ag, norm-conserving PseudoDojo (PDj) pseudopotentials were employed [40]. The exchange-correlation energy was computed using the Perdew, Burke and Ernzerhof (PBE) functional, which is based on generalized gradient approximation (GGA) [41]. For the Brillouin zone sampling, a detailed Monkhorst–Pack k-point grid of 15 × 15 × 15 was used for the optimization of VSe_2_ in its bulk form, while a grid of 15 × 15 × 1 was applied for the monolayer configuration. To mitigate the limitations of the DFT method with regard to accurately representing dispersive forces (van der Waals (vdW) interactions), the Grimme (DFT-D3) corrective dispersion term was incorporated [42,43] and corrective dispersion term was added. The band structure, relative band gap and optical properties were assessed using the Heyd–Scuseria–Ernzerhof (HSE) [44] hybrid functionals. Structural optimization was performed until the residual forces dropped below 0.001 eV/Å. The electronic convergence tolerance was established at 10^−5^ eV. A kinetic energy cutoff of 120 Ry was selected for the integration mesh in all calculations. The Gaussian smearing function was chosen to obtain a smooth metallic band for the electronic structure calculations.

The relative dielectric constant ($\epsilon_{0}(\omega )$) was determined based on Equations (1)–(3) [45]:(1)$$\epsilon_{r}\left(\omega \right)=\left(1+\chi \left(\omega \right)\right)\;,\;$$(2)$$\alpha \left(\omega \right)=V\epsilon_{0}\;\chi \left(\omega \right)\;,\;$$(3)$$\sigma \left(\omega \right)=-i\omega \epsilon_{0}\;\chi \left(\omega \right),\;\;$$
where χ, V and i are the susceptibility tensor, volume and *i-*th component (labeling electrons) of the momentum.

Then, we proposed four nanodiodes of the Cu/VSe_2_ structure using different electrodes of Au and Ag investigated by DFTB [46]. The DFTB method, using its powered and accurate functionals, is taken into account as a principal tool in this field. The earliest [47,48] formalism of DFTB restricted the computations to non-self-consistent interactions between two atoms; later, a second-order approximation of the Kohn–Sham energy was added to consider the charge self-consistent treatment of the systems [49].

SemiEmpirical (SE) approximation [50] applied by Tight-Binding (TB) calculations was used for the simulations of four nanodiodes: Au/VSe_2_/Au, Ag/VSe_2_/Ag, Au/Cu/VSe_2_/Au and Ag/Cu/VSe_2_/Ag. The TB energy $\left(E_{0}^{TB}\right)$ for an M electron system of N nuclei was resolved with Equation (4):(4)$$E_{0}^{TB}=\sum_{i}^{occ}\left\langle \Psi_{i}|{Ĥ}_{0}|\Psi_{i}\right\rangle +E_{rep}\;\;\;\;\;\;$$

This includes occupied Kohn–Sham eigenstates of $\Psi_{i}$, while ${Ĥ}_{0}$ is the Hamiltonian operator, and $E_{rep}$ is rigidly in pairs, repulsive and engaged in short-ranged interactions. To find the solution to the Kohn–Sham equations, the LCAO method is used in an appropriate set of localized atomic orbitals $\varphi_{\nu }$, the single-particle wave functions $\Psi_{i}$, based on Equation (5):(5)$$\Psi_{i}\left(r\right)=\sum_{\nu }C_{\nu i}\varphi_{\nu }\left(r-R_{a}\right)\;\;\;$$
where $R_{a}$ denotes the core distance of ion α. In this approach, we utilize confined atomic orbitals represented in a Slater-type form by addressing an adopted Schrödinger equation of a neutral pseudoatom [51].

In the TB Hamiltonian, the non-self-consistent field (nscf) part was parameterized employing the Muller algorithm, where the elements related to the distance-dependent matrix are described with a numerical operator [52]. The electronical transport characteristics were assessed through the non-equilibrium Green’s function (NEGF) formalism [49,53,54], while it was assumed that coherent electron transport takes place between the source and drain through the central region.

When the self-consistent density matrix was established, I(V) characteristics were assessed by applying the transmission coefficient (*T*) with the electron energy (*E*), as defined by the Landauer equation [55], which is presented in Equation (6):(6)$$I\left(V_{L}T_{L}V_{R}T_{R}\right)=\frac{e}{h}\sum \int T_{\sigma }\left(E\right)\left[f\;\left(\frac{{E-\mu }_{R}}{K_{\beta }T_{R}}\right)-f\left(\frac{{E-\mu }_{L}}{K_{\beta }T_{L}}\right)\right]dE$$
where f is the Fermi energy, and $T_{L}$*/*$T_{R}$ are the electron temperatures of the left/right electrodes, where *L* and *R* electrodes are the source and the drain, respectively. Moreover, $T_{\sigma }\left(E\right)$ is the transmission coefficient for the spin component σ. Finally, the chemical potentials of the right/left electrodes and $V_{bias}$ can be defined by Equations (7)–(10):(7)$$\mu_{R}=E_{F}^{R}-eV_{R}$$(8)$$\mu_{L}=E_{F}^{L}-eV_{L}$$(9)$$\mu_{R}-\mu_{L}=e\;V_{bias}$$(10)$$V_{bias}=V_{L}-V_{R}$$

This methodology indicates that it is possible to obtain a quantitative and computationally efficient representation of coherent transport within solid-state physics. This advancement paves the way for enhanced comprehension and manipulation of charge transport characteristics at atomic-scale interfaces under high bias voltages. In the diode simulations, the periodicity was disrupted in the x and y dimensions, while a constant potential was established along the z-axis utilizing Dirichlet boundary conditions. Dirichlet BCs signify that a certain potential ($V_{0}$) was fixed at the boundary in the S facet in the simulation cell, following Equation (11):(11)$$V^{H}\left(r\right)=V_{0},r\in \;S$$

To evaluate the stabilities of interfaced systems, we calculated the binding energy per atom (*E_B_*) using Equation (12):(12)$$E_{B}=\frac{E(Interface)-E\left(Cu\right)-E({VSe}_{2})}{N}$$
where E(Interface) refers to the total energy for optimized Cu on the ${VSe}_{2}$ surface, $E\left({VSe}_{2}\right)$ and $E\left(Cu\right)$ are the total energies of pristine VSe_2_ and the optimized phase of the Cu atom, respectively, and *N* is the number of atoms in the lattice.

## 4. Conclusions

In summary, a combination of DFT and DFTB computations was carried out to evaluate the material and device features of Cu/VSe_2_ by considering the stability and electronic, optical and electrical responses in the VSe_2_ monolayer and bulk. Two adsorbed systems of Cu/VSe_2_(Top) and Cu/VSe_2_(Corner), were preliminarily simulated with the adsorption of Cu atoms on the surface of the VSe_2_ monolayer, while Cu atoms were positioned on the top and at the corner of the lattice, respectively. Electronic band structures and PDOS calculations showed the zero-band gap of both adsorbed systems, similar to the pure metallic VSe_2_. Optical results indicated that Cu atoms can indeed significantly change the Re(ω) part of the dielectric constant of VSe_2_ under visible, infrared and THz ranges. Moreover, the results related to the visible range revealed that in the zone of the interband/intraband of Cu atoms, higher values of Re(ω) were predicted for both Cu/VSe_2_(Top) and Cu/VSe_2_(Corner), while the negative values of −4.72 (Cu/VSe_2_(Top)) and −4.00 (Cu/VSe_2_(Corner)) were observed at a higher wavelength of 2100 nm (146 THz). The Im(ω) of the dielectric constant indicated that the peak enhancement was from 0.2 THz for Cu@VSe_2_(Top) and 2.9 THz for Cu@VSe_2_(Corner). Refractive index (n) calculations showed increasing refractive indices at 5.4 and 4.6 for Cu/VSe_2_(Top) and Cu@VSe_2_(Corner) along the in-plane (a-) and in-plane (b-) directions, respectively, instead of the corresponding value of 2.9 for the pure system of VSe_2._ Then, four nanodiodes—Au/VSe_2_/Au, Ag/VSe_2_/Ag, Au/Cu/VSe_2_/Au and Ag/Cu/VSe_2_/Ag—were investigated by the DFTB method. While all simulated nanodevices exhibited non-linear I(V) characteristics, Au/Cu/VSe_2_/Au and Ag/Cu/VSe_2_/Ag delivered drastically higher current values at positive and negative voltages than the two other nanodiodes, revealing the effect of the copper layer on the electrical properties of VSe_2_. Overall, our theoretical study demonstrates the prospects of pure and interfaced VSe_2_ nanostructures applied in the THz region.

## Figures and Tables

**Figure 1 ijms-26-02527-f001:**
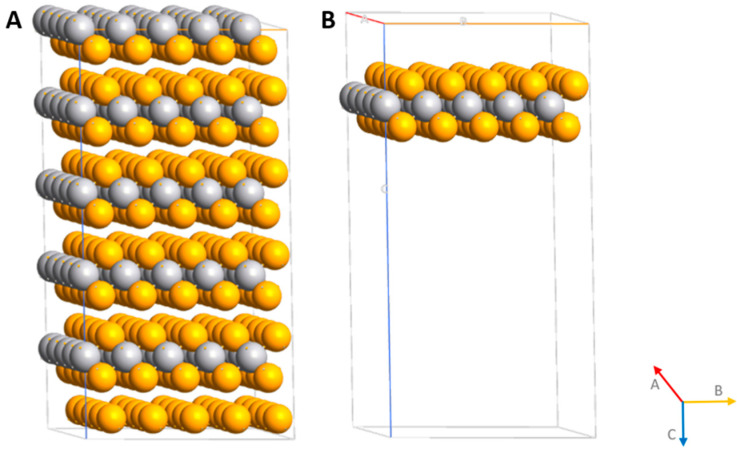
Representation of bulk (**A**) and monolayer (**B**) VSe_2_ modeled in this study. Silver and yellow balls presented the V and Se atoms, respectively.

**Figure 2 ijms-26-02527-f002:**
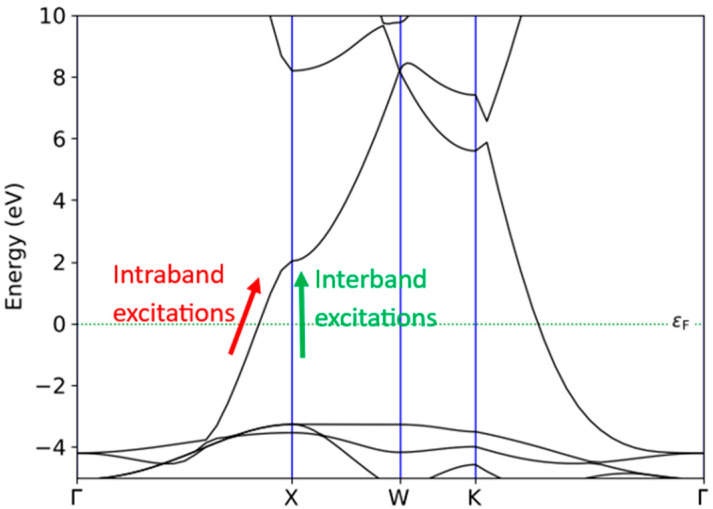
The band structure of metallic Cu computed by the HSE06 functional.

**Figure 3 ijms-26-02527-f003:**
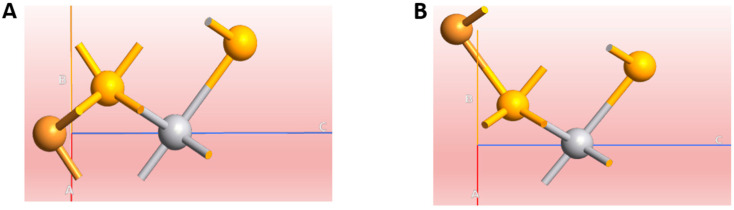
Cu absorption on the VSe_2_ monolayer in two different positions (Top (**A**) and Corner (**B**)). Atoms are colored orange, silver and yellow for Cu, V and Se.

**Figure 4 ijms-26-02527-f004:**
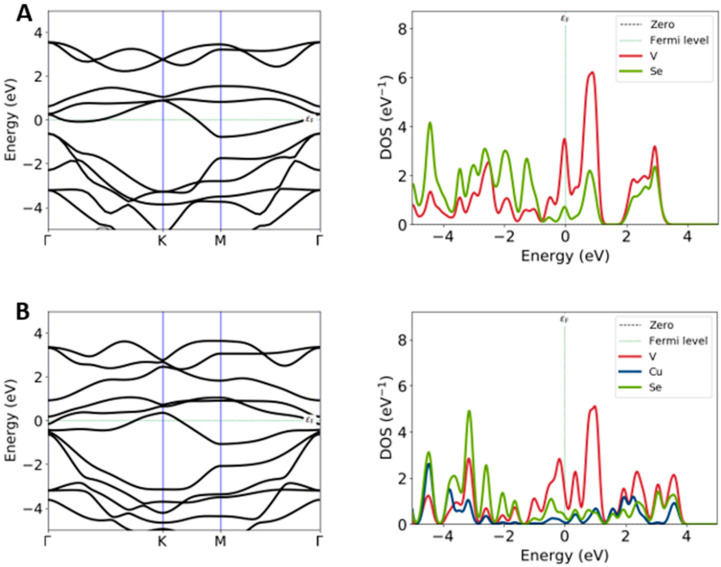
Band structure and PDOS of the VSe_2_ monolayer (**A**) in comparison with the Cu@VSe_2_(Top) interface (**B**).

**Figure 5 ijms-26-02527-f005:**
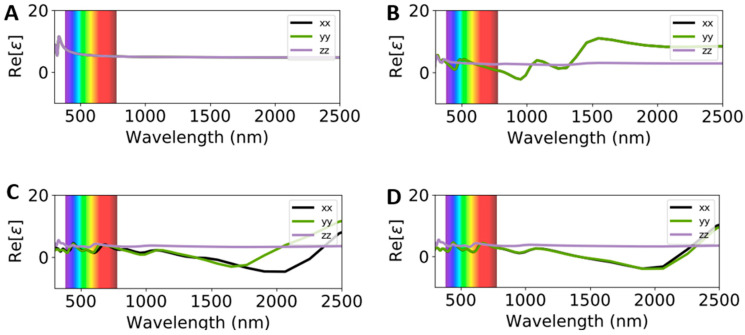
Real of dielectric constants of Cu (**A**), VSe_2_ (**B**), Cu/VSe_2_(Top) (**C**) and Cu/VSe_2_(Corner) (**D**). The color spectrum shows the different wavelengths in the visible range.

**Figure 6 ijms-26-02527-f006:**
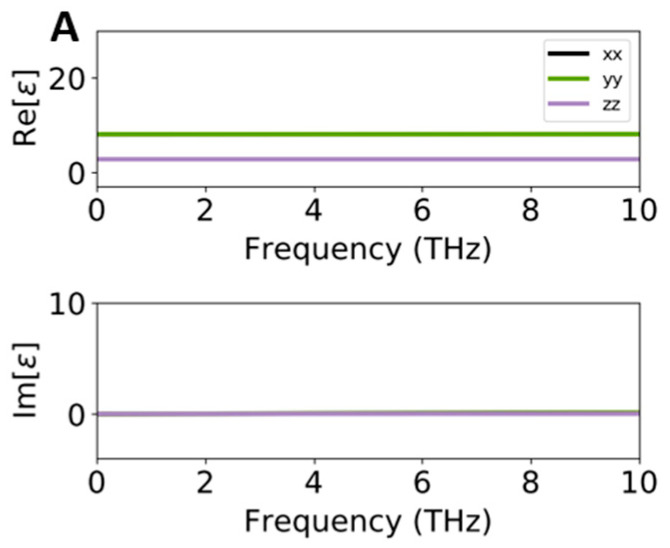
Real and imaginary parts of the dielectric constants of VSe_2_ (**A**), Cu/VSe_2_(Top) (**B**) and Cu/VSe_2_(Corner) (**C**).

**Figure 7 ijms-26-02527-f007:**
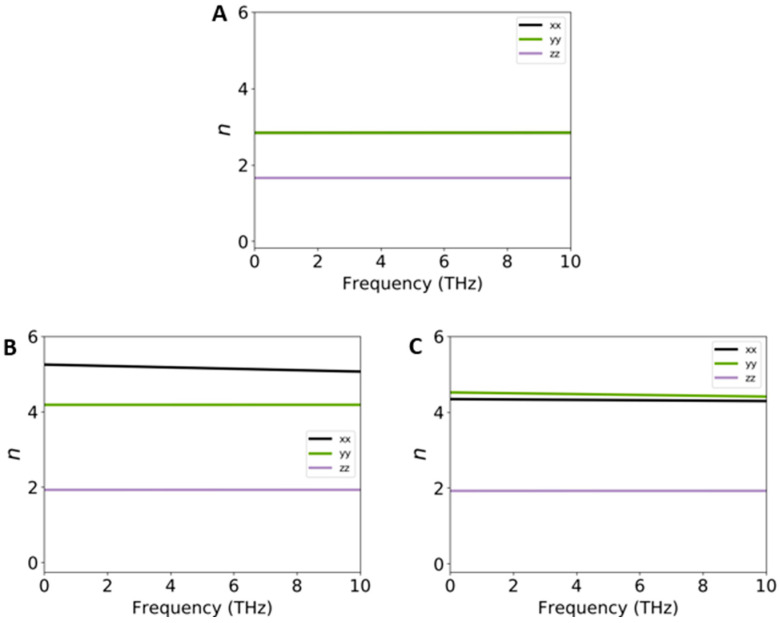
Refractive indexes of VSe_2_ (**A**), Cu/VSe_2_(Top) (**B**) and Cu/VSe_2_(Corner) (**C**).

**Figure 8 ijms-26-02527-f008:**
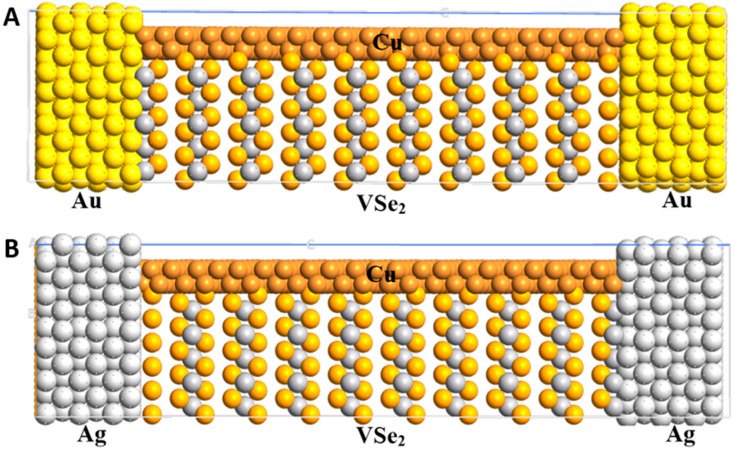
The atomic structure of the Au/Cu/VSe_2_/Au (**A**) and Ag/Cu/VSe_2_/Ag (**B**) nanodiodes.

**Figure 9 ijms-26-02527-f009:**
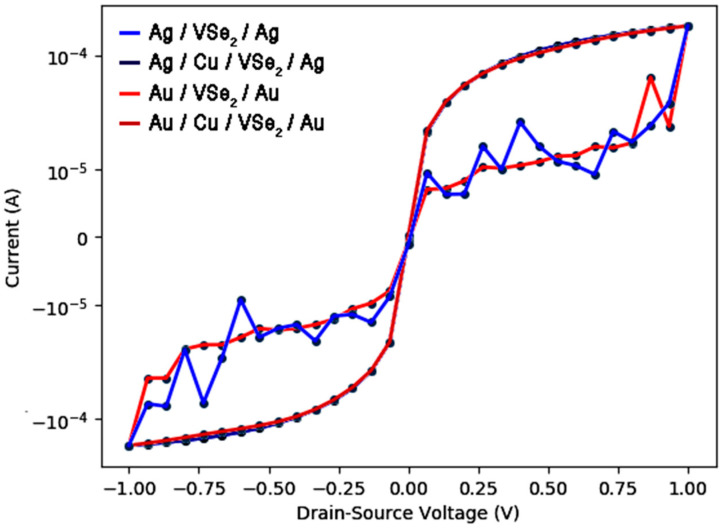
I(V) characteristics of different Au/Cu/VSe_2_/Au, Ag/Cu/VSe_2_/Ag, Au/VSe_2_/Au and Ag/VSe_2_/Ag diodes were modeled in this study.

## Data Availability

No data was used for the research described in the article.

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
