# Peer review of "Tunable Optical Properties of Cu/VSe2 from the Visible to Terahertz Spectral Range: A First-Principles Study"

_ijms, 2025, doi:10.3390/ijms26062527_

Round 1
Reviewer 1 Report
Comments and Suggestions for Authors
In this work, the authors performed DFT and DFTB calculations to analyze the electronic and optical properties of Cu/VSe2 interfaces and VSe2 nanodiodes with and without Cu layers. The results showed that while the metallic nature of VSe2 remained unchanged, the Fermi level peak shifted downward in the Cu/VSe2(Top) configuration. Optical properties revealed enhanced Re(ω) values in the visible range for both Cu/VSe2 configurations, with varying Re(ω) trends in Infrared/terahertz ranges. The presence of Cu increased the intensity of Im(ω) peaks in the dielectric constant and raised the refractive index compared to pure VSe2. DFTB calculations also indicated higher current values in nanodiodes with Cu layers (Au/Cu/VSe2/Au and Ag/Cu/VSe2/Ag) compared to those without Cu.
This work can be interesting for the computational chemistry and the industrial community. I would like to ask the authors to consider the minor comments below.
1. page 3, Figure 2
Two bands show non-smooth behavior around K point between 5 to 7 eV. Can authors provide more discussions on it?
2. page 3, computational details
Reference for the PBE functional is missing: Phys. Rev. Lett. 77, 3865
3. page 3, computational details
The computational details for the smearing need to specified in this section.
4. page 4, line 138
The abbreviation "TB" is used before the definition.
5. page 8, Figure 5
The legend/explanation for the color bars is missing.
6. page 11, Figure 9
Compared to other systems, Ag/VSe2/Ag shows more fluctuations and gives a non-smooth curve, can authors provide more discussions on it?
Author Response
Dear editor and reviewers,
Thanks for your time and efforts devoted to reviewing our paper. We went through the comments and applied them to the manuscript. Also, in this letter, we replied to the comments one by one. To find the changes more conveniently, we coloured them in blue in the main text.
Reviewer #1: In this work, the authors performed DFT and DFTB calculations to analyze the electronic and optical properties of Cu/VSe2 interfaces and VSe2 nanodiodes with and without Cu layers. The results showed that while the metallic nature of VSe2 remained unchanged, the Fermi level peak shifted downward in the Cu/VSe2(Top) configuration. Optical properties revealed enhanced Re(ω) values in the visible range for both Cu/VSe2 configurations, with varying Re(ω) trends in Infrared/terahertz ranges. The presence of Cu increased the intensity of Im(ω) peaks in the dielectric constant and raised the refractive index compared to pure VSe2. DFTB calculations also indicated higher current values in nanodiodes with Cu layers (Au/Cu/VSe2/Au and Ag/Cu/VSe2/Ag) compared to those without Cu.
This work can be interesting for the computational chemistry and the industrial community. I would like to ask the authors to consider the minor comments below.
page 3, Figure 2
Two bands show non-smooth behavior around K point between 5 to 7 eV. Can authors provide more discussions on it?
Respond: Thank you for the carefully comment. Normally, this non-smooth behavior could happen due to several issues in quantum ATK software. The first one is related to discontinuity in the k-path grid, but we chose a high k-points grid of 15 × 15 × 15 for the lattice optimization. We also tested different smearing methods including Fermi-Dirac distribution, Gaussian smearing, MethfesselPaxton smearing, and Cold smearing. Since we are facing with a metal, we used the gaussian smearing, thus more of the k-points were effectively on the fermi surface. On the other hand, we employed very accurate HSE06 hybrid functional, and using other methods like GGA-PBE functional and LDA-PZ, we had the same results in this point. For this reason, we reconsider the band structure output file and the k-points in the reciprocal space of metallic Cu, and the results showed that this is not a simulation error, and non-smooth points are simply related to the “L” point in the first Brillouin zone, in which software skip it because of very close distance to the K point.
- page 3, computational details
Reference for the PBE functional is missing: Phys. Rev. Lett. 77, 3865
Respond: Thank you for the carefully comment. The modification has applied, and we added this reference (no. 36).
- page 3, computational details
The computational details for the smearing need to specified in this section.
Respond: Thank you for this comment, we did not specify out approach in this respective. We used the Gaussian smearing function to obtain a smooth metallic band for the electronic structure calculations. We added these information in the computational section.
- page 4, line 138
The abbreviation "TB" is used before the definition.
Respond: Thank you for this valuable comment. We added the description of TB abbreviation, now the modification can be found in the revised manuscript.
- page 8, Figure 5
The legend/explanation for the color bars is missing.
Respond: Thanks to reviewer for this helpful comment. The explanation of color bars is now added in Figure 5.
- page 11, Figure 9
Compared to other systems, Ag/VSe2/Ag shows more fluctuations and gives a non-smooth curve, can authors provide more discussions on it?
Respond: Thank you for the valuable comment. The I(V) characteristics of Ag/VSe2/Ag diode without presence of Cu atoms showed some noises because of very high resistance behaviour. The resistance fluctuation of noise can observe for some diode samples showing the I(V) characteristic dependent on the electric field orientation (Sita, Zdenek, et al. "Analysis of noise and non-linearity of IV characteristics of positive temperature coefficient chip thermistors." Metrology and measurement systems 20.4 (2013)). Indeed, our theoretical results suggest that the Ag/Cu/VSe2/Ag device with appropriate content of Cu possess abundant electronic trap, which effectively inhibits the recombination of charge carriers, and improving the resistivity in I(V) curves of VSe2. To validate the computational results, such experiments like photovoltage spectroscopy (SPS) or electron spin resonance spectroscopy (ESR) allow to obtain important information on the adsorption nature of Cu atoms on the VSe2 surface, which is out of the scope of this study. We added part of this explanation on the last part of the results and discussion section, together with the reference reported.

Reviewer 2 Report
Comments and Suggestions for Authors
The paper, entitled “Tunable optical properties of Cu/VSe2 from visible to terahertz spectral range: A first-principles study”, submitted for review is an interesting article which investigates computational approaches (DFT and DFTB) to evaluate the electronic and optical properties of different interfaces containing Cu/VSe2. This is a nice contribution towards evaluating the materials and their optical features. The authors present interesting computational results. The article is well written. My opinion is that the topic is of interest to the readership of International Journal of Molecular Sciences so the manuscript is publishable, but after minor revision as follows:
- When determining electronic and optical properties (including luminescence), it is natural to use molecular orbitals - especially to examine HOMO and LUMO orbitals. Why were these parameters not determined in the work? Why was there no determination of the energy gap Egap and the difference in the electronically excited singlet and triplet states ΔEST?
- Technical comments: the article contains stylistic errors and typos that should be corrected before publication; the references are not properly formatted.
Author Response
Dear editor and reviewers,
Thanks for your time and efforts devoted to reviewing our paper. We went through the comments and applied them to the manuscript. Also, in this letter, we replied to the comments one by one. To find the changes more conveniently, we coloured them in blue in the main text.
Reviewer #2: The paper, entitled “Tunable optical properties of Cu/VSe2 from visible to terahertz spectral range: A first-principles study”, submitted for review is an interesting article which investigates computational approaches (DFT and DFTB) to evaluate the electronic and optical properties of different interfaces containing Cu/VSe2. This is a nice contribution towards evaluating the materials and their optical features. The authors present interesting computational results. The article is well written. My opinion is that the topic is of interest to the readership of International Journal of Molecular Sciences so the manuscript is publishable, but after minor revision as follows:
When determining electronic and optical properties (including luminescence), it is natural to use molecular orbitals - especially to examine HOMO and LUMO orbitals. Why were these parameters not determined in the work? Why was there no determination of the energy gap Egap and the difference in the electronically excited singlet and triplet states ΔEST?
Respond: Thank you for this careful comment. Based on our theoretical results in the electronic structure calculations, in both interfaces of Cu@VSe2(Top) and Cu@VSe2(Corner), we found a zero band gap and the only difference in the bandstructure of VSe2 monolayer with Cu/VSe2(interfaces) mostly refer to the band dispersions in the zone of saddle point (SP), in which after the interaction of Cu with VSe2, conduction band shifted to the lower energies. These results reveal that adsorption of Cu on different site of VSe2 cannot effectively change its electronic band diagram. The reviewer is right and the HOMO and LUMO play an important role in how charge carriers (electrons and/or holes) move around Fermi level. However, in this study we have zero band gap (mixing of electron and holes) and there is no energy gap between HOMO and LOMO For this reason, calculation of HOMO-LUMO splitting cannot help us. Indeed, we need to validate our results with other experimental analysis such as Raman spectra, X-ray photoelectron spectroscopy (XPS), X-ray diffraction (XRD), and Laser-induced electron diffraction, to obtain more information about the morphologies of the interfaces. We are working on a subsequent work in which an experimental part will be necessary.
Technical comments: the article contains stylistic errors and typos that should be corrected before publication; the references are not properly formatted.
Respond: The reviewer is right, and we apologize for any confusion. In order to satisfy this comment, we revised again the manuscript to correct any errors and typos, and we reformatted the references.
